# Unravelling the Deformation of Paleoproterozoic Marbles and Zn-Pb Ore Bodies by Combining 3D-Photogeology and Hyperspectral Data (Black Angel Mine, Central West Greenland)

Pierpaolo Guarnieri [1,*], Sam T. Thiele [2] , Nigel Baker [1] , Erik V. Sørensen [1] , Moritz Kirsch [2] , Sandra Lorenz [2] , Diogo Rosa [1], Gabriel Unger [2,3] and Robert Zimmermann [2,4]

1   Department of Mapping and Mineral Resources, Geological Survey of Denmark and Greenland (GEUS), Øster Voldgade 10, 1350 Copenhagen, Denmark; nb@geus.dk (N.B.); evs@geus.dk (E.V.S.); dro@geus.dk (D.R.)
2   Division Exploration Technology, Helmholtz-Institute Freiberg for Resource Technology (HZDR), Chemnitzer Straße 40, 09599 Freiberg, Germany; s.thiele@hzdr.de (S.T.T.); m.kirsch@hzdr.de (M.K.); s.lorenz@hzdr.de (S.L.); g.unger@glu-freiberg.de (G.U.); robert.zimmermann@gub-ing.de (R.Z.)
3   Geologische Landesuntersuchung GmbH Freiberg, Halsbrücker Str. 34, 09599 Freiberg, Germany
4   G.U.B. Ingenieur AG, Halsbrücker Str. 34, 09599 Freiberg, Germany
*   Correspondence: pgua@geus.dk

**Abstract:** The Black Angel Zn-Pb ore deposit is hosted in folded Paleoproterozoic marbles of the Mârmorilik Formation. It is exposed in the southern part of the steep and inaccessible alpine terrain of the Rinkian Orogen, in central West Greenland. Drill-core data integrated with 3D-photogeology and hyperspectral imagery of the rock face allow us to identify stratigraphic units and extract structural information that contains the geological setting of this important deposit. The integrated stratigraphy distinguishes chemical/mineralogical contrast within lithologies dominated by minerals that are difficult to distinguish with the naked eye, with a similar color of dolomitic and scapolite-rich marbles and calcitic, graphite-rich marbles. These results strengthen our understanding of the deformation style in the marbles and allow a subdivision between evaporite-carbonate platform facies and carbonate slope facies. Ore formation appears to have been mainly controlled by stratigraphy, with mineralizing fluids accumulating within permeable carbonate platform facies underneath carbonate slope facies and shales as cap rock. Later, folding and shearing were responsible for the remobilization and improvement of ore grades along the axial planes of shear folds. The contact between dolomitic scapolite-rich and calcitic graphite-rich marbles probably represents a direct stratigraphic marker, recognizable in the drill-cores, to be addressed for further 3D-modeling and exploration in this area.

**Keywords:** 3D-photogeology; hyperspectral data; Greenland; Paleoproterozoic; Zn-Pb ore deposits

## 1. Introduction

The Paleoproterozoic geologic record preserves extensive carbonate platforms and slopes [1] with largely analogous architecture, mineralogy and facies distributions to those present since the Paleozoic [2]. Carbonate platforms are large edifices formed by the accumulation of sediments in areas of subsidence that can be several kilometers thick, extend over many hundreds of square kilometers, and be constructed by stacked carbonate ramps and/or flat-topped platforms [3]. Carbonate slopes encompass a suite of environments which pass seaward from shallow water to the deeper basin-ward settings [3]. In the southern sector of the Rinkian Orogen [4] in central West Greenland (Figure 1), the Mârmorilik Formation [5] includes a major Paleoproterozoic carbonate platform hosting the Black Angel Zn-Pb ore deposit [6]. The lithostratigraphy of the formation is subdivided into a lower dolomite-dominated member and an upper calcitic-dominated member, which are both

intercalated with pelites [7]. The lithostratigraphy was originally established in the relatively weaker deformed South Lakes area (Figure 2) within the South Lake Tectonic Unit [8]. However, at the former mine area at Black Angel Mtn and Akuliarusikassak (Figure 2), more intense deformation and tectonic duplication within the Black Angel Tectonic Unit [8] has altered the primary stratigraphic relationships. This makes it difficult to recognize and interpret original components of the ore-forming mineral system (structures, alteration, etc.) due to the intense deformation and remobilization of the ore [6,9]. However, the presence of anhydrite and of chlorine-rich scapolite in the marbles, interpreted as results of the metamorphism of evaporites, and the general setting of the ore deposit, are considered to support a Mississippi Valley-Type model of mineralization [10].

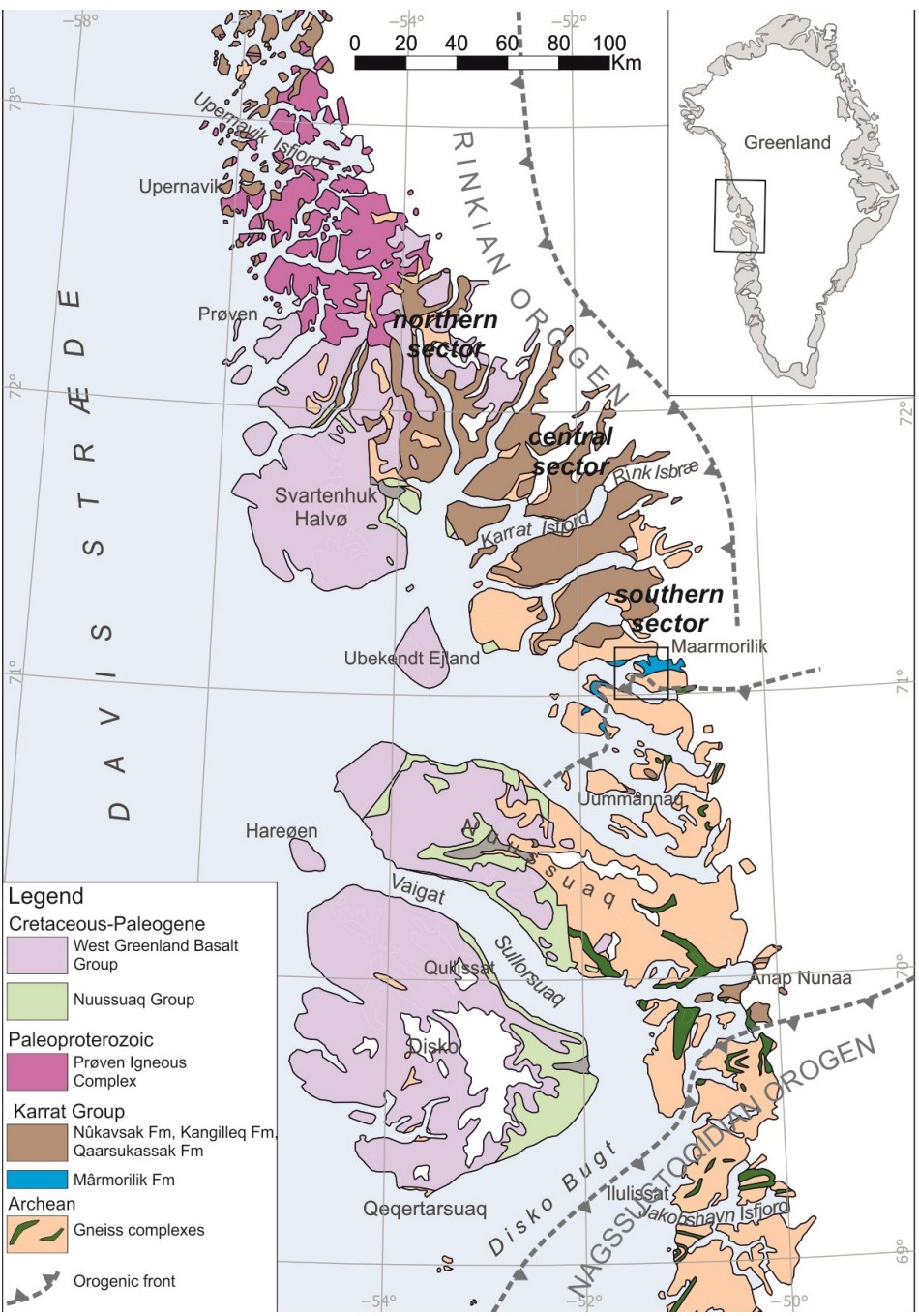

**Figure 1.** Geological map of West Greenland showing the extension of the Rinkian Orogen and the frontal part of the Nagssugtoqidian Orogen (modified after [8]). Black box indicates Figure 2.

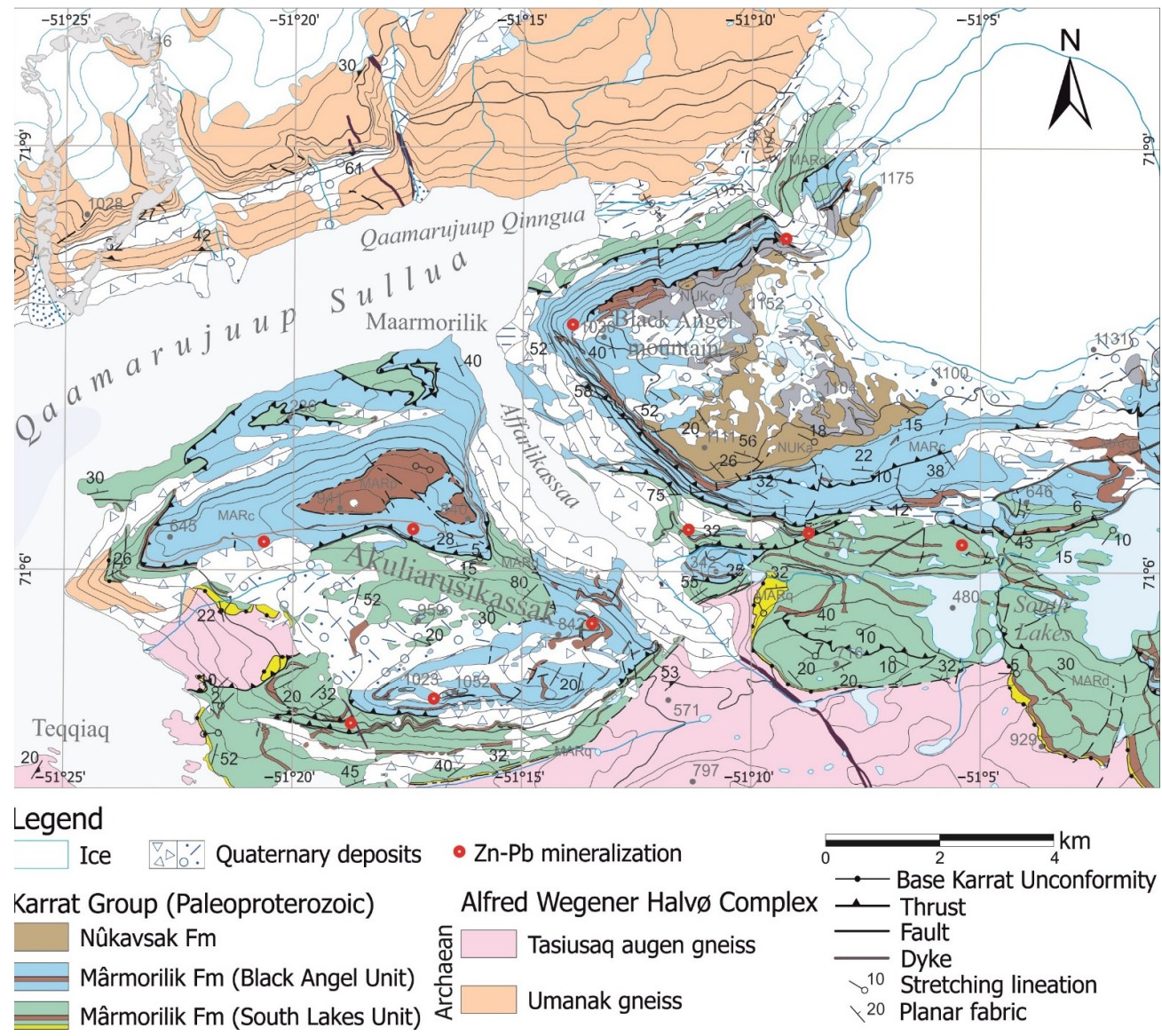

**Figure 2.** Geological map of Maarmorilik (modified after [8]). See location of the area in Figure 1.

The geology of the Black Angel deposit is excellently exposed in the steep and inaccessible cliffs of Black Angel Mountain and Akuliarusikassak and can be readily mapped using 3D-photogeology [8,11]. However, despite the excellent exposures, structural complexity and lateral stratigraphic variations can be difficult to recognize using only RGB images. This limitation can be compensated by using hyperspectral imagery to emphasize lithological or chemical contrasts based on an excellent spectral range. In this contribution we apply, contrast and synthesize a range of innovative virtual outcrop mapping techniques to constrain the lithostratigraphy and structures of the Mârmorilik Formation and Black Angel deposit. These integrate photogrammetric and hyperspectral remote sensing techniques to collect high-resolution data over the largely inaccessible cliffs cutting the Black Angel deposit on the north-east side of Affarlikassaa fjord (Figure 2), offering a near-continuous exposure of a 1200 m high and 4000 m long section through the Mârmorilik Formation.

This integration of hyperspectral data, which can distinguish geologically significant but visually similar minerals such as calcite and dolomite, with 3D-photogeology, which provides an order of magnitude higher spatial resolution, has allowed us to reinterpret fold structures and stratigraphy of the marbles. The result is an integrated stratigraphy showing variations in the calcite vs. dolomite component in marbles and the distribution

of scapolite- and graphite-rich marbles and intercalated organic pelitic layers. A refined structural model explaining the complexity of folds observed in the cliff is presented together with a new stratigraphic evolution for the Mârmorilik Formation, distinguishing carbonate platform deposits which host the mineralization, passing upwards to carbonate slope deposits and shales.

## 2. Regional Geological Setting

The Black Angel Zn-Pb deposit (central West Greenland) is hosted within deformed Paleoproterozoic marbles of the Mârmorilik Formation that belongs to the Karrat Group [5]. The lower part of the Karrat Group is represented by the Qaarsukassak Formation [12] in the central and northern sectors of the Rinkian Orogen and by the Mârmorilik Formation [7] in the southern one (Figure 1). Both formations are overlain by metagreywackes of the Nûkavsak Formation that, in the central part of the orogen, are intercalated with metavolcanic rocks of the Kangilleq Formation (Figure 1). Detrital zircons from the basal quartzites of the Qaarsukassak and Mârmorilik formations, suggest that the Karrat Basin initiated somewhere after ca. 2000 Ma. The lower units formed in an intracratonic sag basin as the Rae Craton subsided. The basin progressively evolved through an intracratonic rift stage, with associated volcanism and syn-rift siliciclastic sedimentation between 1950 and 1900 Ma [13], to a later stage back-arc geodynamic setting [4].

Between 1900–1850 Ma, the northern part of the Karrat basin was intruded by felsic plutons of the Prøven Igneous Complex (PIC) [4,14] and subsequently affected by the Rinkian metamorphism between 1830 1and 800 Ma [4,15], which reached the upper greenschist facies in the south (Maarmorilik) and granulite facies in the north [4]. The Rinkian orogen is interpreted as part of the Trans-Hudson Orogeny (St-Onge et al., 2009) and represents a back-arc fold and thrust system, resulting from the east-ward collision between the magmatic arc (PIC) and the Karrat basin [4]. The inversion of the Karrat Basin involved a first stage of thin-skinned tectonics, with allochthonous metagreywackes emplaced during east-verging thrusting, followed by thick-skinned tectonics with progressive involvement of the Archaean gneisses with basement nappes and metasediments transported toward the North-East [8,16].

Rinkian structures in the Maarmorilik sector were later overprinted by NW-SE compression with NE-SW trending folds and thrusts associated with the external front of the Nagssugtoqidian Orogen (Figure 1) [8,16].

### 2.1. The Mârmorilik Formation

The tectonostratigraphic subdivision of the Mârmorilik Formation is described in detail by Guarnieri et al. [13], based on the presence of tectonic discontinuities and sheared contacts. The Nunngarut Thrust separates the succession into two tectonic units: the South Lakes Tectonic Unit in the footwall and the Black Angel Tectonic Unit in the hanging wall. The tectonic units are roughly equivalent to the dolomitic and calcitic members defined by Garde [7] but juxtaposed by the Nunngarut Thrust. This implies that the overall thickness of the Mârmorilik Formation should be considered the result of tectonic juxtaposition of the deeply folded Black Angel Tectonic Unit onto the largely undeformed South Lakes Tectonic Unit, so does not represent the original stratigraphic thickness of the Formation.

The base of the South Lakes Tectonic Unit preserves a depositional contact between Archaean basement rocks and quartzites displaying symmetrical wave ripples, containing a monomictic metaconglomerate [7]. The succession continues with variegated dolomitic marbles and minor calcitic marbles with pelitic beds, intercalated at different stratigraphic intervals. The latter, indicating a siliciclastic input to the carbonate platform after c. 1915 Ma [13].

In the hanging wall of the Nunngarut Thrust, the dominantly calcitic marbles of the Black Angel Tectonic Unit contain phlogopite, <20% dolomite, minor quartz, albite and graphite and occasionally scapolite [7]. The latter mineral indicates the presence of evaporites [5,10] prior to metamorphism. The Black Angel Tectonic Unit also comprises

banded calcitic-dolomitic marbles and the intercalated pelites show an increase in thickness upwards [5,13].

### 2.2. Black Angel Mountain and Zn-Pb Ore Deposit

The Black Angel Zn-Pb deposit is hosted by marbles of the Mârmorilik Formation and associated pelites, whose shape in outcrop resembles a "Black Angel" and give their name to the 1150 m high plateau mountain hosting the deposit (Figure 3). This strata-bound deposit consists of massive to semi-massive sphalerite-galena-pyrite, and is hosted in folded, anhydritic marbles. The deposit encompasses eight different ore bodies and two satellite ore bodies in Akuliarusikassak (the Nunngarut ore bodies) (Figure 2). The Black Angel mine hosted pre-mining reserves of 13 Mt, at 12% Zn, 4% Pb and 29 ppm Ag [17] and was active between 1973 and 1990, with 11.2 Mt of ore extracted during this period. It has been classified as an exhalative sediment-hosted massive sulfide (SEDEX) deposit [18,19] and as an epigenetic Mississippi Valley-type (MVT) deposit [20–22], while Horn et al. [23] favored a metamorphic Kipushi-style model. Recently, the authors in [10] interpreted the Black Angel deposit as evaporite-related Mississippi Valley-Type deposit, in which the overlying organic-rich semipelites and massive calcitic marbles may have served as seals for hydrocarbons and reduced sulfur to chemically react with mineralizing fluids and deposit the sulfidic ore.

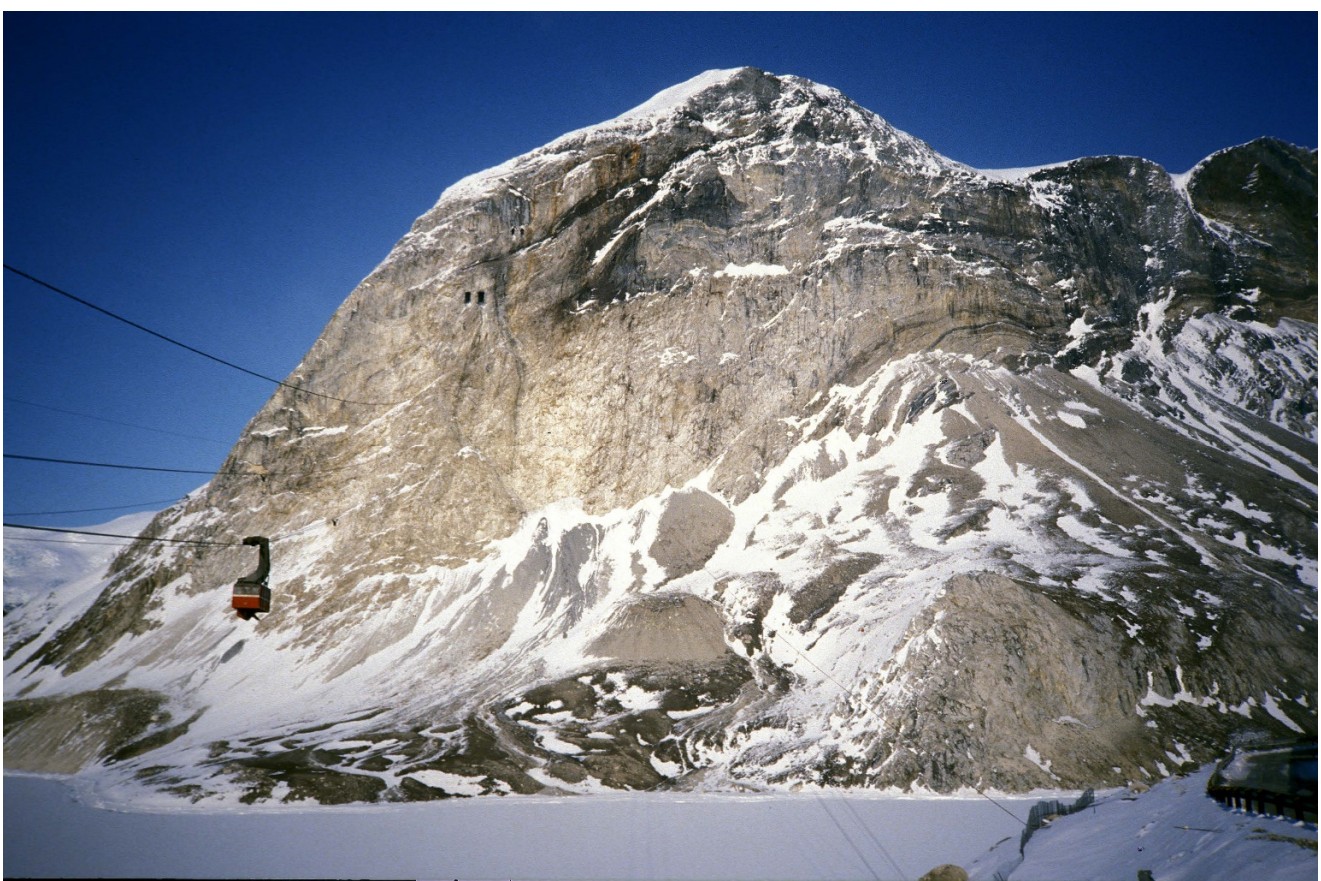

**Figure 3.** Historical photo of the Black Angel mine. Note the cable-car going toward the mine entrances at ca. 600 m elevation, close to angel-like shape of the black pelites. The Maarmorilik mine town is located at the bottom right of the picture.

## 3. Methodology

### 3.1. 3D-Photogeology

Photogrammetry is a classical remote sensing technique that allows geologists to make three-dimensional observations of geological outcrops from collections of two-dimensional

images acquired from different locations. Today, digital photogrammetry is an efficient and powerful geological tool used for geological problem solving [24] within the geoscientific community. Examples are plentiful and range several orders of scale, including detailed close-up investigation of ice surfaces [25], to near real-time volcanic eruption monitoring [26] and regional geological map making [13,27,28].

We use oblique images collected during field work in 2015 in the Maarmorilik area (Figure 4). These are a subset of a larger collection of more than 50,000 oblique images [11] collected across the Rinkian orogen by GEUS.

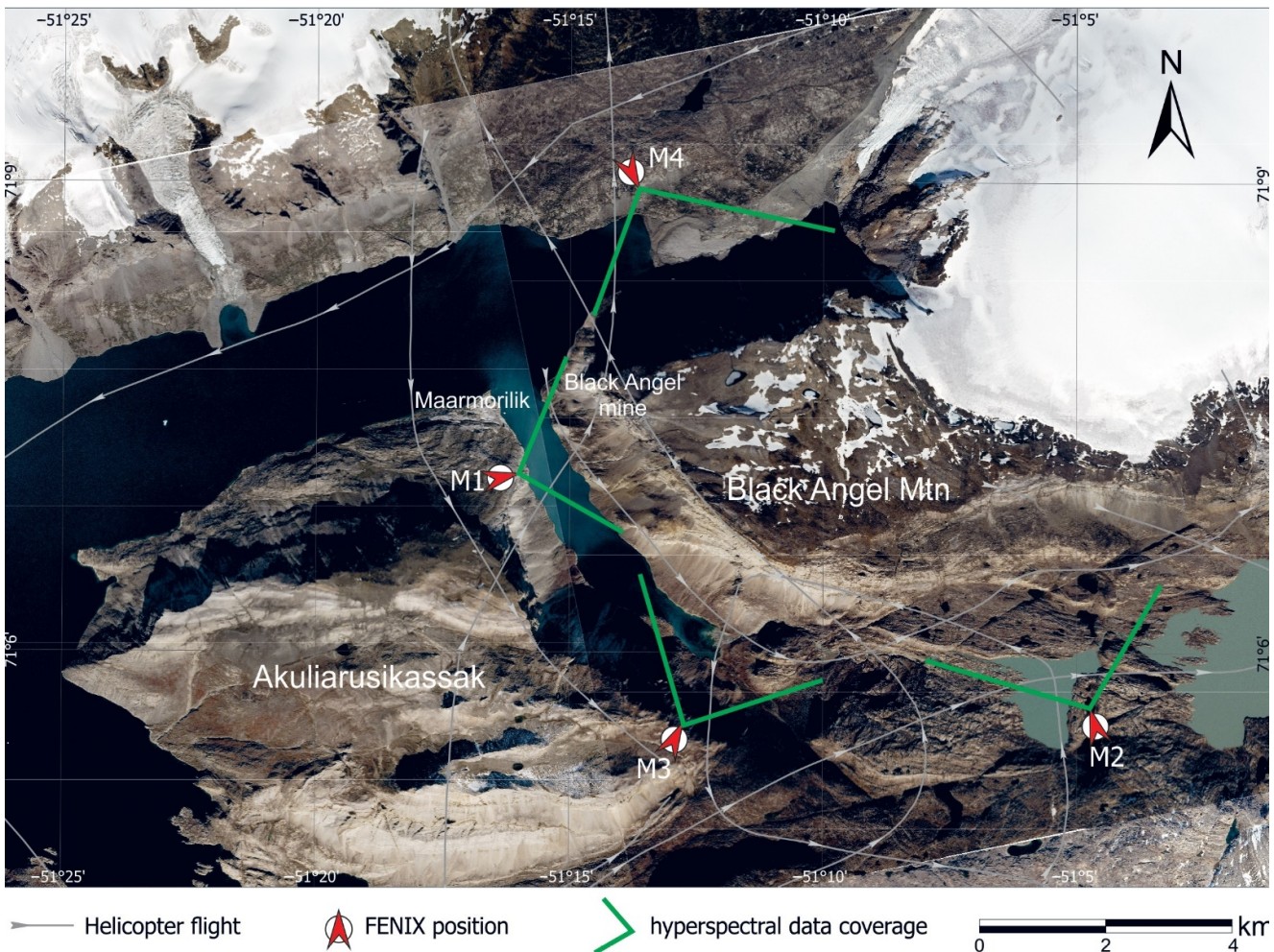

**Figure 4.** Helicopter flightlines and scanning positions of the hyperspectral camera showing the related data coverage of the investigation area.

The images were collected using a Nikon D800E (36 megapixel) digital, single lens reflex camera equipped with a Carl Zeiss Distagon 35 mm f1.4 lens. The camera system (camera and lens), which was focused and kept fixed at infinity during data acquisition, was calibrated prior to field using a calibration test grid. It was deployed on a helicopter and operated in a handheld and manual fashion through an open helicopter window. The study area was covered by several photographic flybys at different distances. Overview flight lines, typically covering the entirety of the cliff, have a Ground Sampling Distance (GSD) of c. 0.4 m (pixel size on the ground), while close-up images have a GSD in the 0.1–0.2 m range and were targeted to specific regions of interest. The images were generally acquired with 90% overlap and were prepared for geological interpretation following the workflow outlined in [24]. We used a combination of global navigation satellite system (GNSS) data, ground control points collected in 1:150,000 monochrome aero-triangulated images and sea

level levelling points to calculate the orientation of the images using Anchor Lab's 3D Stereo Blend software. Three-dimensional Stereo Blend was subsequently used to map the geology in three dimensions. The result of this is a set of 3D polylines that follow intersection traces between the outcrop surface and lithological contacts, identified by their color differences. Built-in tools within 3D Stereo Blend allowed easy semi-automated geologic cross-section generation, which was edited and finalized with digital editing software.

### 3.2. Hyperspectral Data

Hyperspectral data are increasingly being fused with photogrammetric virtual outcrop data to add additional information on mineralogy and distinguish otherwise cryptic lithologies ([29–32]). This approach has been particularly successful for mapping carbonate minerals that are otherwise difficult to distinguish (e.g., calcite and dolomite) [29]. Hence, a portable hyperspectral sensor (Specim AsiaFENIX) was used to acquire visible-near (VNIR; 430–1000 nm) to short-wave (SWIR; 1000–2500 nm) infrared hyperspectral data covering the Black Angel Mtn cliff during August 2016 (Figure 4).

These hyperspectral data and associated correction routines are described in detail in the accompanying data publication [33]. This workflow followed the method for illumination correction and 3-D fusion described by Thiele et al. [34,35]. Path radiance effects caused by the large (1–3 km) viewing distances were corrected using the method of Lorenz et al. [36]. The resulting reflectance hypercloud has a spatial resolution of ca. 3 m and was processed using band ratio and minimum wavelength mapping techniques [37,38] to distinguish the dominant carbonate minerals, which are associated with characteristic absorption positions in the shortwave infrared range [39].

## 4. Integrated Stratigraphy of the Black Angel Cliff Section

### 4.1. Drill-Core Stratigraphy

Logging from six drill-cores (V3, V5, V17, 1814, 1570 and 1572 in Figure 5a), which were drilled by Cominco in between 1966 and 1984, have been used to constrain the stratigraphy proximal to the Black Angel deposit. These holes were selected as they are located within 300 m of the cliff face, allowing comparison with our surface observations. For the purpose of this paper, the various logged lithologies are divided into five groups following a scheme established by the mine geologists in company reports [40]:

(1) White Massive marble: massive, white to off-white and medium to coarse grained anhydrite-bearing marble.

(2) Light-Grey Banded marble: Light-grey scapolite-bearing marble weakly foliated with light brown mica. Disseminated grey to pink fluorite is often conspicuous, defining a weak color banding.

(3) Dark-Grey Banded Marble: Distinctly banded to well-banded marbles, fine- to medium grained with at least two distinct subunits (a) grey, fine- to medium grained, well banded containing chert fragments (chert-bearing marble) and (b) light grey to grey, fine-grained and diffusely banded marbles hosting the ore sulfides (ore-bearing unit).

(4) Grey Banded Marble: Massive to weakly foliated Grey to dark grey, fine grained, marble with graphite. Exclusively seen as a transitional unit between marble and pelite units and interpreted to represent finely interbedded carbonate and pelite.

(5) Pelite: Massive to finely laminated black, fine grained pelite often inter-banded with fine-grained, dark grey marble on mm- to several cm-scale. Pyrite and rare pyrrhotite is reported.

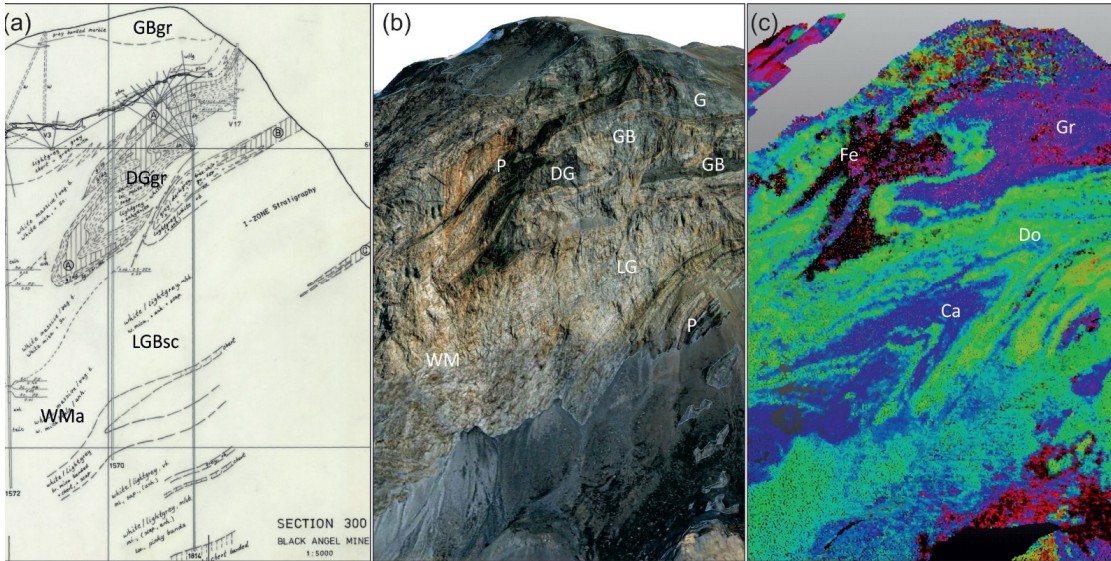

**Figure 5.** Integrated stratigraphy. (**a**) Drill-core stratigraphy from the NE-SW oriented Section 300 of Cominco database (after van der Stijl 1984): WMa: White Massive Marble + anhydrite; LGBsc: Light-Grey Banded Marble + scapolite; DGgr: Dark-Grey Marble + graphite; GBgr: Grey-Banded Marble + graphite; P: Pelite/chert; (**b**) RGB stratigraphy: WM: White Massive Marble; G: Grey Marble; GB: Grey Banded Marble; LG: Light-Grey Marble; DG: Dark-Grey Marble; P: Pelite; (**c**) Hyperspectral stratigraphy: Ca: Calcite-rich; Do: Dolomite-rich; Fe: Iron-rich; Gr: Graphite-rich.

### 4.2. Hyperspectral Stratigraphy

The presence of intimate mixtures of, e.g., calcite, dolomite and tremolite in the marbles of the Mârmorilik Formation make the accurate identification of mineral abundance from remotely sensed hyperspectral spectra a significant challenge [33]. These complexities are further compounded by the presence of various lichen species, which can be abundant in the Maarmorilik area and sometimes exhibit absorption features in the 2180–2300 nm range [41].

In spite of these challenges, the band ratio and minimum wavelength mapping that are described in depth in Lorenz et al. [33] reveal several distinct lithological contacts that, while difficult to confidently interpret with respect to mineralogy, reveal a variety of fold and fault structures within the Mârmorilik Formation (Figure 6). The geometry of these is inconsistent with patterns that might be caused by, e.g., variable lichen cover, so must relate to variations in the abundance of calcite, dolomite and tremolite in the deformed marbles.

### 4.3. RGB-Stratigraphy and Structures

In the oblique photos of Black Angel Mountain, it is possible to distinguish white-massive marbles from light-grey marbles or dark-grey marbles from banded marbles and the pelites, based on color (Figure 5b). The ore body is located within light-grey banded marbles along the axial plane of the "Z-fold" defining the Black Angel structure (Figure 7a), which formed due to parasitic folding in the core of a larger synform dominated by white-massive marbles. The synform plunges 150° toward the NE, while the main structural trends (fold axes and stretching lineations) are oriented NW-SE to WNW-ESE [8]. According to Guarnieri and Baker [8], the Black Angel Tectonic Unit is a thrust sheet in the hanging wall of the Nunngarut Thrust separated from the South Lakes Tectonic Unit in the footwall. The overall structure is better described as a stack of thrusts where the Mallak Thrust (MaT, out of the section) represents the trailing edge of the thrust system [8]. The lower limb of the synform hosting the Z-fold is, in turn, sheared along the Black Angel Thrust (BaT) and the white calcitic marbles belonging to this unit appear to be tectonically repeated (Figure 7b).

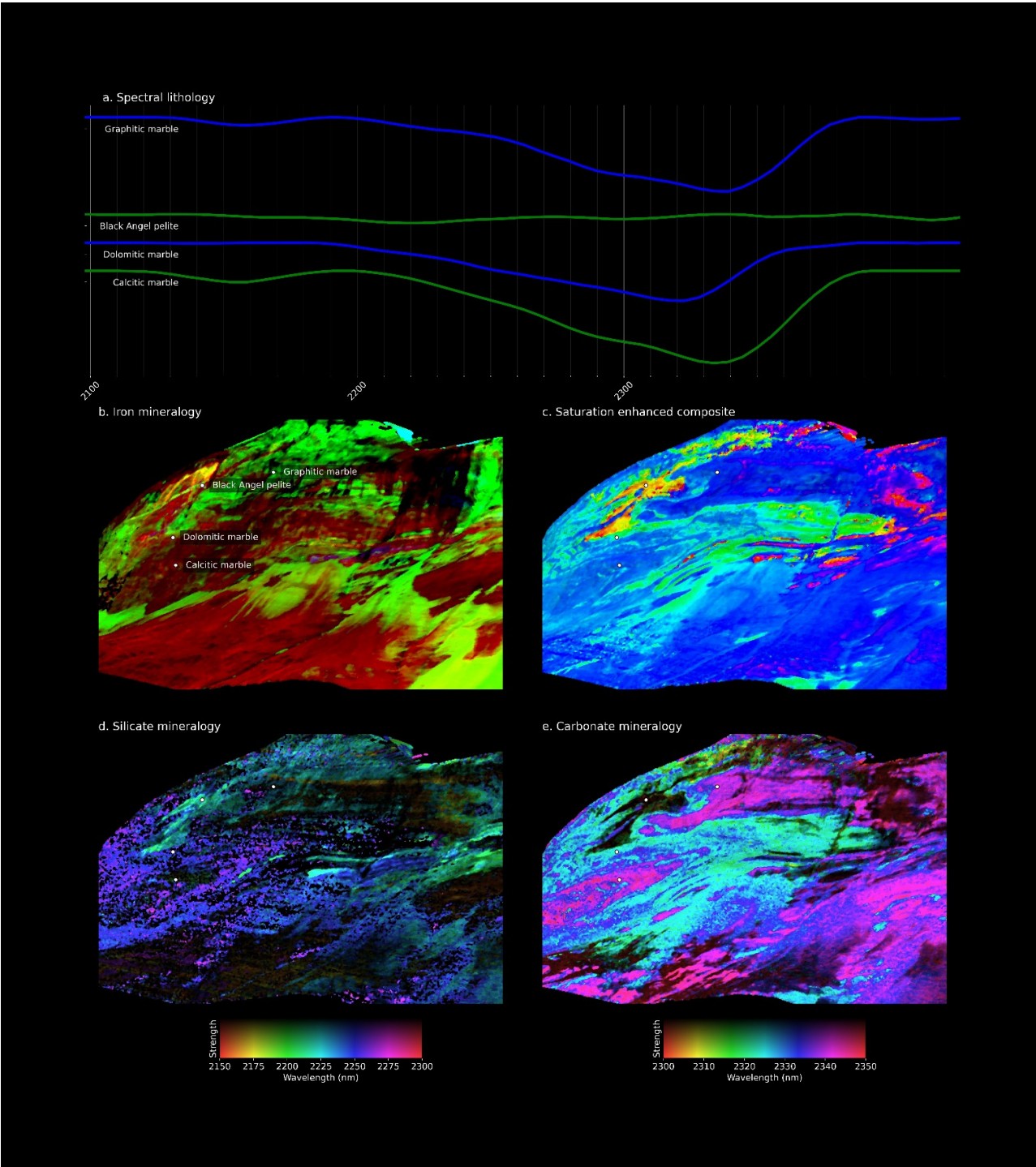

**Figure 6.** View of the main Maarmorilik cliff looking NE-wards and showing the main hyperspectral visualization products used to identify and map stratigraphic horizons. The shortwave infrared contains several diagnostic absorption features allowing the characterization and discrimination of many of the lithologies in the Marmorilik formation (**a**). These allow qualitative mapping of iron mineralogy (**b**) using standard band-ratio approaches [33] and the identification of otherwise cryptic stratigraphic horizons using false-color visualizations of spectral absorbance at 2200, 2250 and 2350 nm (**c**). More detailed insights can be gained using minimum wavelength mapping techniques to identify silicate (**d**) minerals containing Al-OH (e.g., white mica and clay minerals; green to cyan) and Fe-OH (e.g., amphibole, chlorite; blue to purple), or distinguish (**e**) calcitic (purple) and dolomitic (cyan) marbles.

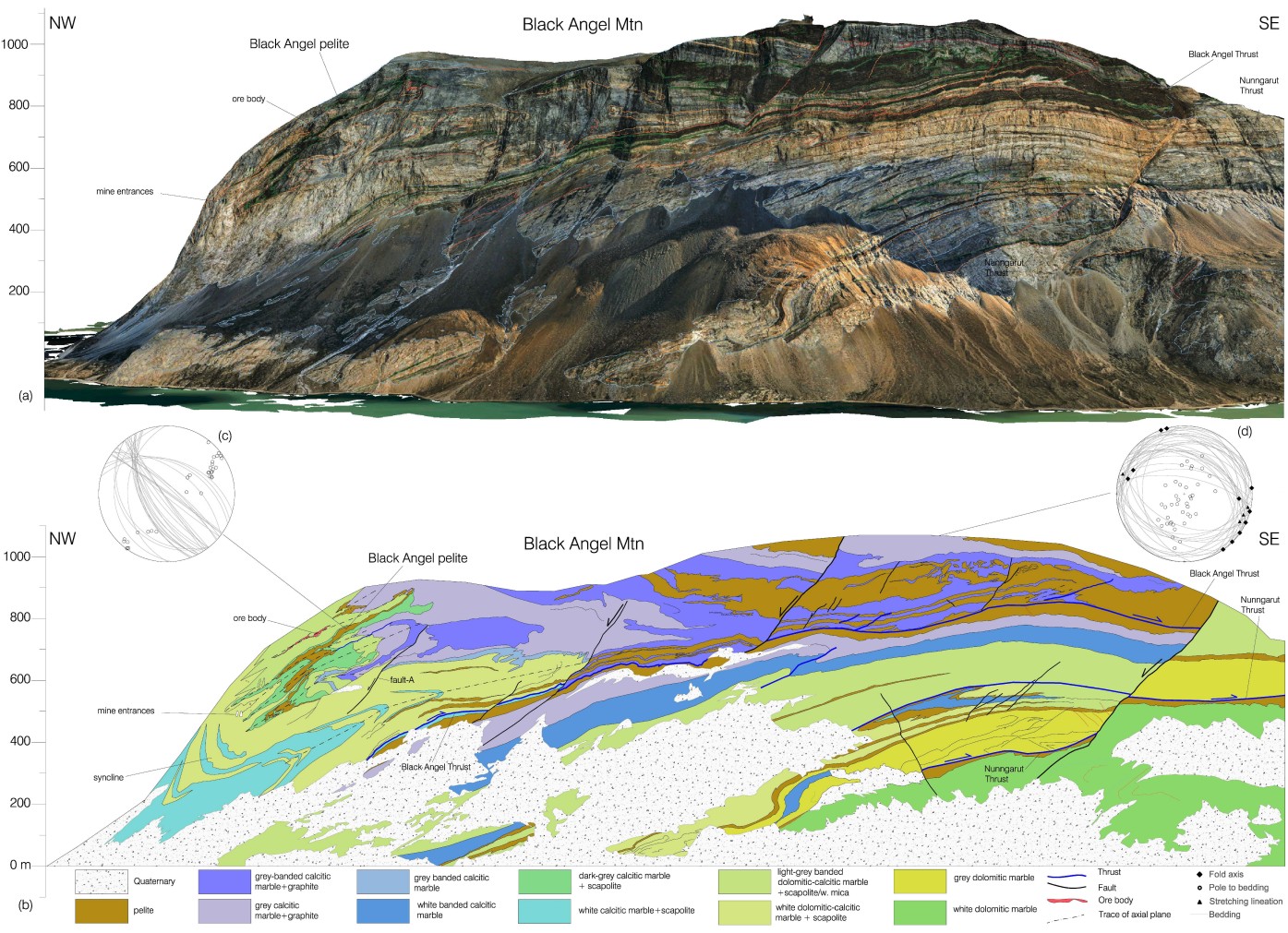

**Figure 7.** Comparison between Virtual Outcrop Model (**a**) and the geological map of the Black Angel cliff (**b**) obtained from the integrated stratigraphy and showing major structures mapped from 3D-photogeology and different mineral composition of the marbles (see legend in Figure 5). (**c**) lower hemisphere stereoplot of planar fabric from 3D-photogeology and (**d**) planar and linear fabric collected in the field (modified after [8]).

## 5. Discussion

The most striking features highlighted by the integrated stratigraphy are (1) the geometry of folds within the white massive calcitic marbles, (2) the detailed stratigraphic relationships above and below the ore body and associated Black Angel pelite, and (3) the lateral distribution of graphite within marbles. Based on the new features, it is possible to define the style of deformation that controlled the ore remobilization and to reconstruct the stratigraphic setting that controlled the ore formation (Figure 7b).

### 5.1. Fault-Propagation Folds and Ore Remobilization

The presence of a syncline at the bottom left of the cliff (Figure 7b) was based on interpretation of the geometry of the Black Angel pelite located above it [8]. The intercalations between dolomitic and calcitic marbles highlighted by the hyperspectral stratigraphy clearly show a coupled syncline-anticline that was not previously recognized (Figure 7b). The core of this anticline consists of white scapolite-rich calcitic marbles, passing upwards to white scapolite-rich dolomitic-calcitic marbles, and the inclined to recumbent fold has the lower limb sheared along the Black Angel Thrust (Figure 7). Similar shear folds are also observed within grey-banded calcitic marbles in the hanging wall of the Black Angel thrust at the top right of Figure 7b. The fold geometries and their structural relationship with thrusts in the NW corner of Black Angel Mtn can be interpreted as transported fault propagation folds [42] related to an imbricate thrust system (Figure 8).

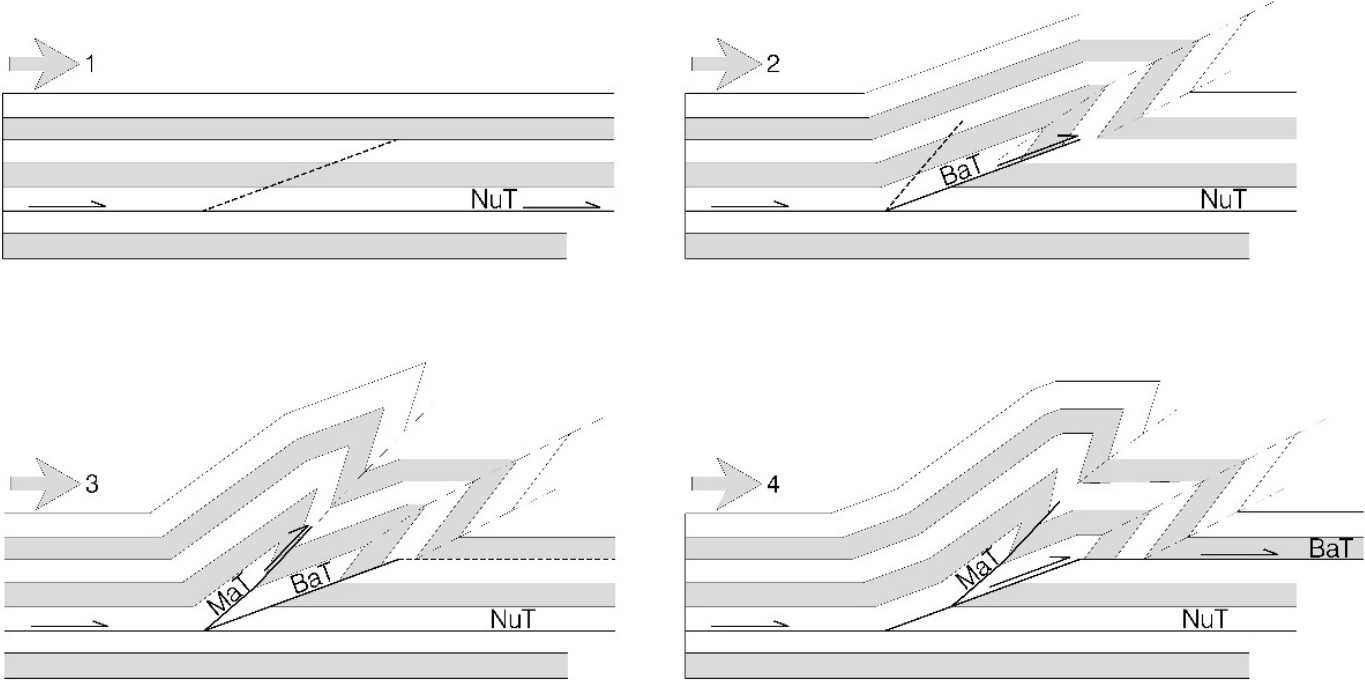

**Figure 8.** Cartoon showing the progressive deformation of the Black Angel tectonic unit interpreted as fault-propagation fold. NuT-Nunngarut Thrust, BaT-Black Angel Thrust, MaT-Mallak Thrust.

The imbricate thrust is generated by propagation of the thrust tip along a ramp of undeformed strata (Black Angel and Mallak thrusts) and then transported by a thrust that has broken through onto an upper flat (Black Angel Thrust). The Black Angel pelite thus forms a parasitic Z-fold in the lower limb of a larger syncline in the footwall of the Mallak Thrust that, together with the Black Angel and Nunngarut thrusts, represent an imbricate thrust stack (Figure 8d). The ESE-trending axial planes of the folded marbles and pelite are tilted more than 60° towards the SW, establishing that the folds are related to simple shear passive folding with a SW-ward direction of tectonic transport. The latter agrees with the NW-SE trends of fold axes obtained from oblique images (Figure 7c) and with WNW-ESE

trending stretching lineations and fold axes collected in the field on top of Black Angel Mountain (Figure 7d).

The timing for the thrust stacking is not constrained by geochronology data but presumably postdates the Rinkian metamorphism that is dated 1830–1800 Ma [4]. The ore body was probably remobilized along the axial plane of the parasitic fold [8] during this time.

### 5.2. An Updated Stratigraphic Model

The data presented in this paper support the study in [7], regarding the general stratigraphy and the interpretation that calcitic and dolomitic marbles form large laterally extensive layers that are conformable with clastic/pelitic marker horizons. This implies that dolomitization occurred prior to metamorphism and deformation. Using the integrated stratigraphy (Figure 7b), a conceptual restoration of the folded Black Angel pelite was developed (Figure 9). Although the fold limbs are slightly sheared or boudinaged, the original stratigraphic relationships seem to be preserved as no major offsets are observed.

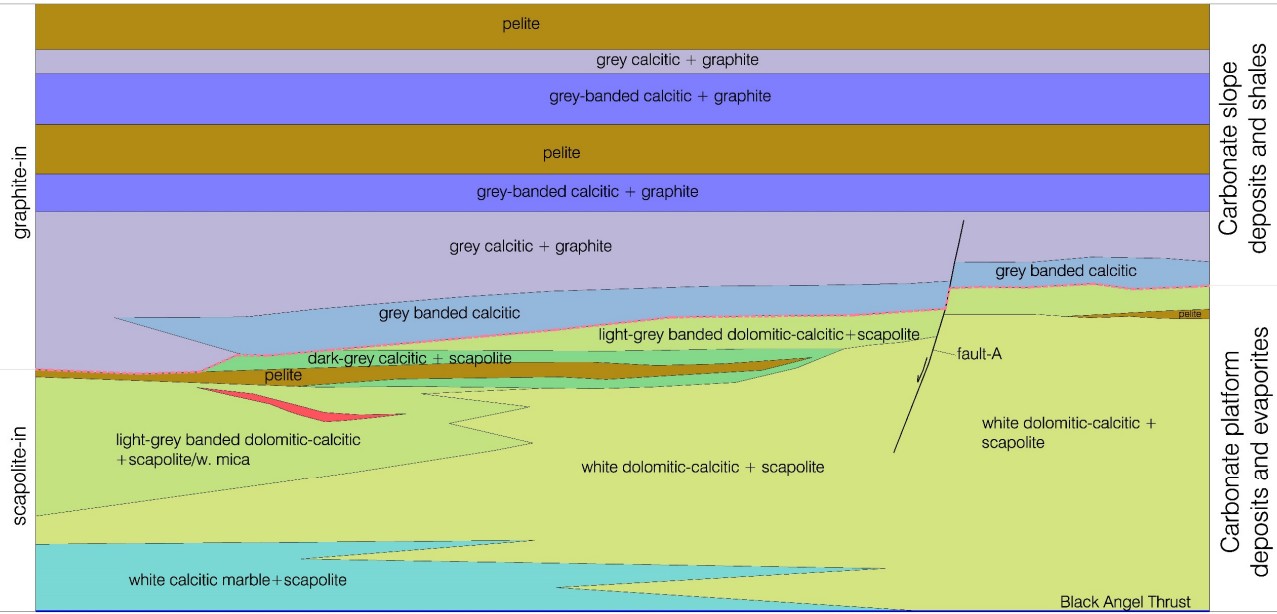

**Figure 9.** Stratigraphic chart of the folded Black Angel pelites after restoration of ca. 1800 m wide and 600 m thick rock package, showing the relationships of the stratigraphic units above and below the pelites (not to scale). Red color corresponds to the ore deposit; legend as in Figure 7.

The reconstructed stratigraphic chart (Figure 9) refers to the area around the Black Angel pelite and extends south-east of fault-A (Figure 7b). The data show scapolite-rich marbles in the lower part and graphite-rich marbles in the upper part (Figure 9). The iron index (Figure 6b) suggests the presence of oxidized iron in the dolomitic marbles and reduced conditions in the graphitic marbles. The scapolite-rich marbles are mostly dolomitic, apart from the white calcitic marbles at the bottom, and probably represent the carbonate platform facies deposits. In the upper package, the graphite-rich marbles are always calcitic and are intercalated with thick pelites, probably represent the carbonate slope facies deposits and shales. The Black Angel pelite, which is intercalated with c. 7 m dark-grey scapolite-rich calcitic marbles, is interpreted as maximum flooding surface that closely coincides with the graphite/scapolite boundary (Figure 9) and marks the drowning of the platform. Finally, the ore body is hosted within light-grey scapolite-rich dolomitic-calcitic marbles, immediately below the c. 10 m-thick Black Angel pelite. This updated stratigraphic model for the Mârmorilik Formation can be summarized as a superposition

of two different paleoenvironments: a succession of evaporite-carbonate platform deposits, followed by carbonate slope deposits and shales (Figure 10).

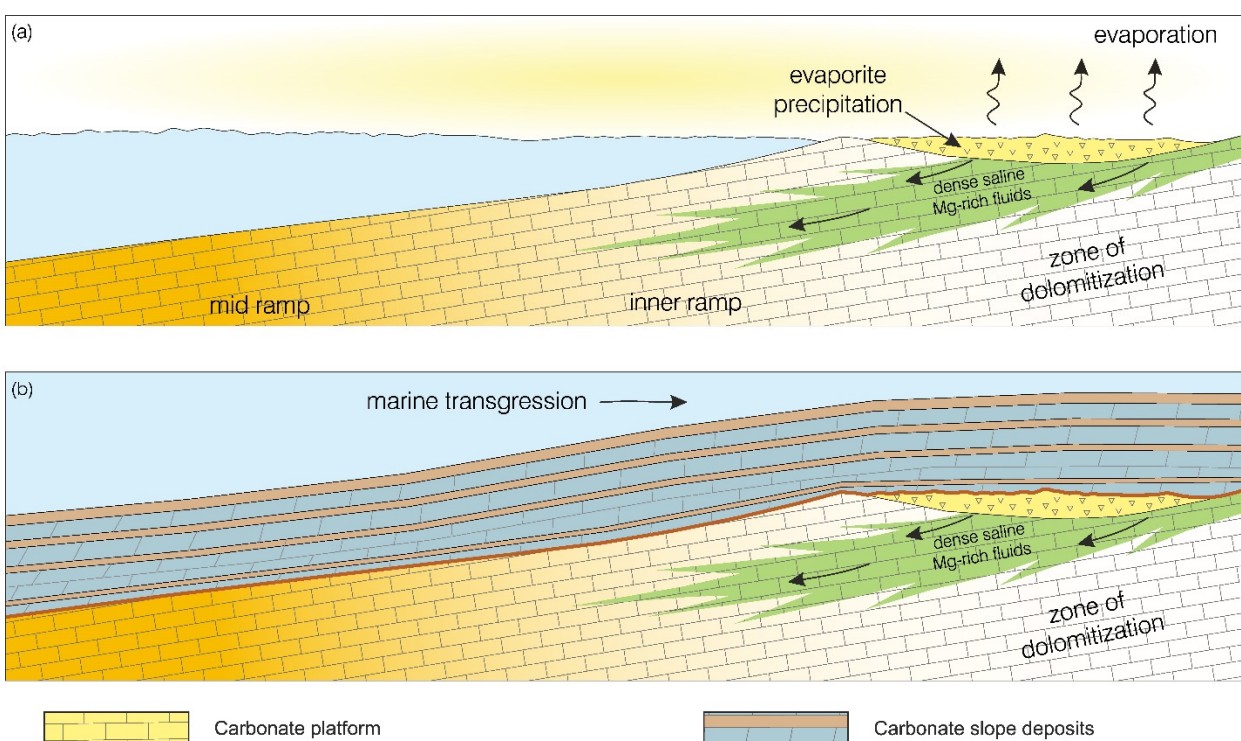

**Figure 10.** Stratigraphic evolution of the Mârmorilik Formation from (**a**) evaporite-carbonate platform facies to (**b**) carbonate slope facies and shales.

The evaporite-carbonate platform includes stromatolites, reported in both drill-cores [43] and outcrop [22], suggesting a biogenic origin for the carbonate platform and a shallow water or intertidal depositional environment for the dolomitic marbles. Moreover, the prismatic, cm-sized crystals of chlorine-rich scapolite found in the dolomitic marbles [7,10] may represent pre-metamorphic evaporitic minerals [44,45], which is further indicative of a shallow water depositional environment. The distribution of strontium in the marbles indicates that the calcitic marble was not formed by metasomatic de-dolomitization of dolomitic precursor but originated from a calcitic precursor [46]. Therefore, it may be assumed that the carbonate in the evaporite-carbonate platform was originally deposited as calcite and the formation of dolomite resulted from exposure to Mg-rich fluids during diagenesis at shallow depth. In this environment, the formation of dolomite can be explained by a process analogous to that operating today in supratidal coastal sabkhas [47], where diagenetic dolomite forms by replacement of aragonite due to the anomalously high salinity and Mg/Ca ratio of shallow pore fluids through seepage reflux (Figure 10a) [47,48].

The carbonate slope deposits above the carbonate platform succession are represented by grey-banded calcitic marbles that contain up to 1% graphite [7] within pelite-rich horizons (Figure 7). The absence of dolomitization suggests this succession was never exposed to a shallow supratidal coastal sabkha environment. Instead, pelite units must have been deposited in deeper water, suggesting carbonate slope sedimentation and shales associated with a deeper water environment (Figure 10b).

## 6. Conclusions

Excellent exposures along the inaccessible alpine landscape of central West Greenland represent a perfect target for photogrammetry and 3D-photogeology. However, RGB

images alone can struggle to identify chemical/mineralogical contrasts, particularly in lithologies dominated by minerals with a similar color (e.g., marbles).

This paper shows how the integration of RGB data with hyperspectral imagery enhanced our ability to identify stratigraphic units and extract associated structural information. The resulting map strengthens our understanding of deformation style in the marbles by identifying previously unrecognized fault propagation folding, and indicates a paleoenvironmental subdivision within marbles of the Mârmorilik Formation. These results suggest that the Black Angel ore formation was likely controlled by two factors:

1. The stratigraphy: ore is confined within marbles of the carbonate platform (reservoir) below the carbonate slope and shales (cap rock);
2. The structures: folding and shearing are responsible for the remobilization and upgrading of the ore along the axial planes of shear folds. Early structures (potentially including Fault A; Figures 7 and 9) could also have played an important role.

In conclusion, further modeling for mineral exploration in the Black Angel Mtn should address the stratigraphic contact between graphite-rich calcitic and scapolite-rich dolomitic marbles, probably the easiest stratigraphic marker recognizable in drill-cores, together with the WNW-ESE trend of the structures. Our interpretation of the dolomite as diagenetic also indicates that it can be used as an indication of stratigraphic position and a coastal sabkha-like depositional environment.

**Author Contributions:** Conceptualization, P.G.; methodology, P.G., N.B., E.V.S., S.T.T., M.K. and S.L.; software, E.V.S. and S.T.T.; HSI data acquisition and pre-processing, R.Z.; formal analysis of hyperspectral data, S.T.T., R.Z. and G.U.; visualization, E.V.S., N.B. and S.T.T.; writing—original draft, P.G.; writing—review and editing, P.G., M.K., S.L., E.V.S., N.B., D.R., G.U. and R.Z. All authors have read and agreed to the published version of the manuscript.

**Funding:** This research is co-funded by the Ministry of Mineral Resources of Greenland (MMR).

**Data Availability Statement:** Not applicable.

**Acknowledgments:** Field work carried out within the framework of the Karrat-Zinc project, jointly financed by the Geological Survey of Denmark and Greenland (GEUS) and the Ministry of Mineral Resources of Greenland (MMR).

**Conflicts of Interest:** The authors declare no conflict of interest.

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
