# Peer review of "Unravelling the Deformation of Paleoproterozoic Marbles and Zn-Pb Ore Bodies by Combining 3D-Photogeology and Hyperspectral Data (Black Angel Mine, Central West Greenland)"

_minerals, doi:10.3390/min12070800_

Round 1
Reviewer 1 Report
The authors have developed a new workflow for recognition of structure and lithology, which can be used in future geological exploration and prospecting. The work holds significant practical meaning and have relatively complete processes, so, here I address some suggestions rather than revision comment.
1. Using a drone can be much cheaper than using a helicopter, and drones can carry multiple kinds of cameras, including hyperspectral and infra-red ones. Related 3D modeling softwares are also quite useful, like the Pix4D mapper, smart 3D etc. All these can reduce your workload and costs. Perhaps you can discuss this in the Discussion.
2. In China, we are developing geological education software platforms for college students. In order to gather enough data, we use drones to scan geological sections, ore pit and other field things. Then, unreal engine is used to build an interactive system to simulate field work, including walking, running, measuring exposed sections and lithology identification (naked eyes). Do you have similar plans for college education? This may extend the Conclusion and assign your research more practical significance.
Author Response
- We use the helicopter because we always have the helicopter as a platform for fieldwork. Moreover, the outcrops' size (in some cases 20-30 km long stretches) is not suitable for drones;
- We do develop Virtual Outcrop Models that can be used for education, but that is not this paper's topic.
Reviewer 2 Report
The manuscript describes a combined application of 3D photogrammetry with hyperspectral analysis on a restricted access cliff in Greenland. The techniques allow the recognition of fold and fault structures with high resolution, as well as the identification of different phases of carbonates and sulfates, thus leading to a new stratigraphic reclassification of the Mârmorilik sequence locally, with a reconstruction of paleoenvironments. The text is well written, organized, and richly illustrated. The results obtained would be even more interesting if they were used as a basis for an integrated 3D geological model of the Black Angel deposit. Just as a suggestion, if photos of Zn-Pb sulfide ore were added, mainly illustrating the remobilization of the ore along axial planes of shear folds, it would enrich the research work.
Saulo de Oliveira
Author Response
There are two new papers recently published (Guarnieri and Baker, 2022 Journal of Structural Geology; Rosa et al., 2022 Mineralium Deposita) dealing with structures and ore re-mobilization (with photos of Zn-Pb ore), to which we refer in the manuscript.
Reviewer 3 Report
The manuscript addresses the very interesting subject of 3D photogeology and hyperspectral remote imaging in identifying the geology of remote and difficult to access areas, and its correlation to ore bodies.
The manuscript is well written, with well-described methodology, results and discussion. Only minor corrections need to be addressed, mainly for the unfamiliar to the area readers.
Please refer to the attached revised version of the manuscript for details.

Author Response
the manuscript's text of the updated version is revised according to reviewer #3, in red colour.
-Caption of Figure 1 is modified
-Figure 2 is modified
-Figure 4 is modified
-Figure 7 is modified